# Prediction of Honeydew Contaminations on Cotton Samples by In-Line UV Hyperspectral Imaging

**DOI:** 10.3390/s23010319

**Published:** 2022-12-28

**Authors:** Mohammad Al Ktash, Mona Stefanakis, Frank Wackenhut, Volker Jehle, Edwin Ostertag, Karsten Rebner, Marc Brecht

**Affiliations:** 1Center of Process Analysis and Technology (PA&T), School of Life Sciences, Reutlingen University, Alteburgstraße 150, 72762 Reutlingen, Germany; 2Institute of Physical and Theoretical Chemistry, Eberhard Karls University Tübingen, Auf der Morgenstelle 18, 72076 Tübingen, Germany; 3Texoversum Faculty Textile, Reutlingen University, Alteburgstraße 150, 72762 Reutlingen, Germany

**Keywords:** hyperspectral imaging, pushbroom, UV spectroscopy, principal component analysis, PCA, partial least squares regression, PLS-R, discriminant analysis, DA, cotton, sugar, honeydew

## Abstract

UV hyperspectral imaging (225 nm–410 nm) was used to identify and quantify the honeydew content of real cotton samples. Honeydew contamination causes losses of millions of dollars annually. This study presents the implementation and application of UV hyperspectral imaging as a non-destructive, high-resolution, and fast imaging modality. For this novel approach, a reference sample set, which consists of sugar and protein solutions that were adapted to honeydew, was set-up. In total, 21 samples with different amounts of added sugars/proteins were measured to calculate multivariate models at each pixel of a hyperspectral image to predict and classify the amount of sugar and honeydew. The principal component analysis models (PCA) enabled a general differentiation between different concentrations of sugar and honeydew. A partial least squares regression (PLS-R) model was built based on the cotton samples soaked in different sugar and protein concentrations. The result showed a reliable performance with *R*^2^_cv_ = 0.80 and low RMSECV = 0.01 g for the validation. The PLS-R reference model was able to predict the honeydew content laterally resolved in grams on real cotton samples for each pixel with light, strong, and very strong honeydew contaminations. Therefore, inline UV hyperspectral imaging combined with chemometric models can be an effective tool in the future for the quality control of industrial processing of cotton fibers.

## 1. Introduction

Hyperspectral imaging is an imaging technology that combines image analysis with spectroscopy [1,2]. Precisely, it is a series of images acquired by moving the object or the imager. It is a fast and non-destructive technique, which has developed into a robust analysis tool for product screening. Such systems are able to capture spectral and spatial information with high resolution. As a result, a spectrum of an object can be obtained for each pixel simultaneously [3,4].

Hyperspectral imaging and spectroscopic applications are widely used in industrial environments [5,6,7,8,9,10]. The importance of hyperspectral imaging is steadily growing in the textile industry. For example, the visible (Vis) and near-infrared (NIR) ranges are often applied for quality control and sorting processes [11,12], where the UV range is rarely used so far.

In textile research, cotton is considered as one of the most important natural fibers for fabric production [13,14]. It provides approximately 50% of the world’s textile fibers [15]. Cotton consists of approximately 95% cellulose and 5% sugar, wax, proteins, organic acids, and pectin [16]. The process ability is affected and degraded by the sugar content. Sugar is a natural excretion of aphids and whiteflies on the cotton through metabolic processes and is specifically called honeydew [17]. This contamination on the raw cotton causes stickiness, which causes problems in the processing stage. This leads to economic loss because the sticky raw cotton is rejected during quality control [16,18]. Inline detection and subsequent removal of sticky cotton would lead to an uninterrupted production and, thus, higher profit [12,19,20].

Several detection methods have been developed and applied to identify cotton contaminations and stickiness in ultraviolet (UV)-Vis/NIR spectroscopic applications as well as Vis/NIR hyperspectral imaging [2,21,22,23]. Most of them are offline such as optical spectroscopy [23,24]. Identification of cotton and cotton trash components was studied by Fortier et al. [25] using FT-NIR spectroscopy. Mustafic et al. [26] examined the applicability of hyperspectral imaging to detect and classify cotton foreign matter in the visible spectral region, whereas the Vis/NIR region was studied by Jiang et al. [27]. Other methods, such as thermogravimetric analysis [28], high-pressure liquid chromatography (HPLC), and the Minicard [29], require an elaborate sample preparation, and are time-consuming and expensive compared to optical spectroscopy. Inline detection and quantification of the stickiness on cotton samples using NIR hyperspectral images were investigated by Severino et al. [20]. They were able to discriminate between glucose on cellulose and melezitose, trehalose, glucose, fructose, and sucrose at each pixel.

Tschannerl et al. [30] compared hyperspectral imaging in UV and NIR regions to precisely discriminate between phenolic flavor concentrations in melted barley. The rarely used UV region showed interesting results despite the illumination not being optimal. Previously, our group reported a hyperspectral imaging setup for the UV spectral region, where this setup was used to distinguish between different pharmaceutical drugs [5], as well as for characterizing the oxide layers’ thickness and copper states on direct bonded copper [6]. The results clearly showed that a spectral imager based on pushbroom technology has many advantages in terms of achieving short UV operational wavelengths and a high spectral resolution. However, hyperspectral imaging rapidly scans samples, resulting in a large amount of spectral data within a short time period. Therefore, multivariate data analysis, such as principal component analysis (PCA) and partial squares regression (PLS-R), is required to reduce the amount of data without losing important information. PCA reveals the similarities and the differences among the samples of a data matrix [31]. A combination between PCA and quadratic discriminant analysis (QDA) enables data classification and investigation of model quality parameters [32,33]. The information about the relation between a number of predictor variables and independent variables can be extracted by using PLS-R [34].

The aim of this work was to develop a chemometric model able to identify and quantify the amount of honeydew on real cotton samples based on UV hyperspectral imaging. For this approach, a reference sample set, which consists of honeydew typical sugars and proteins, was prepared. Mechanically cleaned cotton was soaked with solutions with different sugar concentrations. Chemometric models, especially PCA and PLS-R, were developed based on UV hyperspectral imaging. PCA was used to classify the cotton samples according to their sugar concentration and PLS-R was applied to correlate the UV spectra with the sugar concentration. This PLS-R model successfully predicted the amount of honeydew in grams on real cotton samples. This work is considered as the first scientific work to identify and quantify the amount of honeydew content at each pixel. Therefore, hyperspectral imaging is a suitable technique for inline environment applications in a rapid and non-destructive manner.

## 2. Materials and Methods

### 2.1. Chemicals and Preparation of Solutions

The sugar and protein solution that was applied on the cotton sample was prepared based on the components of real honeydew [35,36,37]. An amount of 0.2 g of each macronutrient 1–6 was weighted and dissolved in 10 mL of deionized water (Table 1). A sixfold serial dilution was prepared in 50 mL volumetric flasks. For each diluting step, 25 mL of the previous solution and 25 mL of deionized water were mixed for 2 min (Table 2).

### 2.2. Sample Set and Sample Preparation

The sample set consisted of cleaned cotton where different amounts of sugar and protein solutions were added to build the model and real cotton samples orginally contaminated by honeydew to test the model.

The cotton samples for model building were collected from a bulk of cotton mechanically cleaned [38,39] from the Texoversum Faculty Textile at Reutlingen University. The cotton was a blend of different long staple Pima qualities. In total, 21 cotton samples were prepared with a weight of 0.3 g ± 0.0001 g (XSE205 DualRange, Mettler Toledo GmbH, Switzerland). The samples were dried in a vacuum oven (Vacutherm VT 6130 M, Termo Fisher Scientific Inc., Waltham, MA, USA) at 30 °C and 50 mbar for 8 h to remove absorbed humidity. The humidity was estimated by a commercially available sensor (Humidity-Detector MD, H. Brennenstuhl GmbH & Co. KG, Tübingen, Germany). The weight loss is documented in Appendix A. An amount of 4 mL of macronutrient solution was used for each sample. Three samples per concentration were made (Table 2). The samples were soaked in an aluminum plate (28 mL, Carl Roth GmbH + Co. KG, Karlsruhe, Germany). The samples were dried again in a vacuum oven at 30 °C and 50 mbar for 44 h. By determining the remaining weight, the average macronutrient content can be calculated for each sample (Appendix A). The humidity and temperature of the laboratory were monitored (BL30, Klima-Datenlogger, Trotec GmbH, Heinsberg, Germany) during the whole workflow.

Comparable real cotton samples were collected by ICA Bremen GmbH (Bremen, Germany) to test the predictive power of the model. The samples were chosen according to their honeydew content in the steps light, strong, and very strong [40]. The samples origin were Sudan Acala (Table 3). The samples were measured and classified by ICA Bremen GmbH company. The measurement was based on different single points to count their stickiness due to honeydew contamination. Therefore, the samples with less sticky points present light samples. Strong and very strong samples have more sticky points (Table 3).

Figure 1 shows the samples pressed in the sample holder prepared for measuring. The sample types were named from A to F and one mechanically cleaned (CLN) sample, where A had the highest concentration (2 wt%) and F had the lowest concentration (0.0625 wt%) (Table 2). The average sugar content remaining on the samples after 44 h was calculated (Table 2). For ease of reading, the term macronutrients was omitted for the description of the solution of various sugars and the protein in the following, and it was replaced with the short-term “sugar” for the sample nomenclature.

### 2.3. UV Hyperspectral Imaging Setup

Figure 2a shows a scheme of the hyperspectral imaging setup. The pushbroom imager consists of a back-illuminated CCD camera (Apogee Alta F47: Compact, inno-spec GmbH, Nürnberg, Germany) and a spectrograph (RS 50-1938, inno-spec GmbH, Nürnberg, Germany) with a slit width of 30 µm. The CCD camera has a resolution of 1024 × 1024 pixels (spatial × spectral) and a pixel size of 13 µm × 13 µm. The optimal integration time was 300 ms. The conveyor belt (700 mm × 215 mm × 60 mm, Dobot Magician, Shenzhen Yuejiang Technology Co., Ltd., Shenzhen, China) moves with a constant speed of 0.15 mm/s. The conveyor belt was located totally in a polytetrafluoroethylene (PTFE) (Sphereoptics GmbH, Herrsching, Germany) tunnel. The illumination was achieved by two Xenon lamps (XBO, 14 V, 75 W, OSRAM, München, Germany). Figure 2b shows a sample holder developed to reduce the influence of the topography of the samples. Therefore, a quartz glass made of suprasil 2 grade B with the dimensions of 140 mm × 80 mm × 1 mm (Aachener Quarzglas-Technologie Heinrich GmbH & Co.KG, Aachen, Germany) was used. PTFE was used as a white reference.

Figure 2c–e illustrate the principle and workflow of the data acquisition. Figure 2c presents the principle of the hyperspectral imaging line scanning method, which collects one line at a time, with all of the pixels in a line being measured simultaneously. The continuous line-by-line collection of spectral information results in a lateral (*x*, *y*) 2D image, as shown in Figure 2d, whereas each location contains a further spectroscopic dimension (*λ*), as shown in Figure 2e. Thus, a 3D data matrix (hypercube) was recorded.

### 2.4. Data Collection and Preprocessing

The UV hyperspectral imaging data were acquired by the SI-Cap-GB version V3.3.x.0 software (inno-spec GmbH, Nürnberg, Germany). The reflectance was calculated by the SI-Cap-GB automatically after recording I_reference_ and I_dark_. Illumination conditions will vary between samples and even within samples across the scan line, especially for heterogeneous samples such as cotton with high scattering due to sample topography. A common method to reduce the influence of the sample topography is to convert the raw spectra of each pixel into reflectance spectra (radiometric calibration) using the following formula [5,31,41,42,43]:(1)$$\mathrm{Reflectance}={\frac{R}{R}}_{0}=\frac{I_{\mathrm{sample}}\;-\;I_{\mathrm{dark}}}{I_{\mathrm{reference}}\;-\;I_{\mathrm{dark}}}$$

R and R_0_ are the intensities reflected from the sample and a specific reference material with high reflectivity, respectively, in this case, PTFE. The intensity of the original image is represented as I_sample_. Accordingly, the intensity of the dark current image is given by I_dark_ and the intensity of the PTFE image is I_reference_ [4]. In order to enhance the absorption bands, the negative decadic logarithm is calculated as −log (R/R_0_).

Hyperspectral data matrices were analyzed by Evince version 2.7.13 (Prediktera AB, Tvistevägen, Sweden). It is used for data handling and extracting the spectra of each pixel. The sensor size is 1024 pixels × 1024 pixels. This means 1024 pixels are in the *x*-direction (lateral resolution in *x*) and 1024 pixels are the wavelength *λ* from 225 nm to 410 nm (variables/columns in the PCA matrix). By moving the conveyor belt line by line, 4896 lines were recorded (lateral resolution in *y*). For model building, the sample set contained 4896 lines × 1024 pixels, resulting in approximately 5 million spectra (rows in the PCA matrix). The dimensions of the PCA were 5 million rows (objects/spectra) × 1024 columns (variables/wavenumbers). With 4464 lines × 1024 pixels, which represent approximately 4.6 million spectra, the model was tested. In total, this resulted in a data size of approximately 15 GB. Figure 3a shows an example of false-colored hyperspectral images at 290 nm of cotton samples sprayed with different concentrations of sugar solution. A region of interest was selected by using a rectangular shape to extract the spectra (Figure 3b).

### 2.5. Multivariate Data Analysis and Model Building

Multivariate data analysis (MVA) was performed with “Aspen Unscrambler^TM^, version 10.5.1” (Aspen Technology Inc., Bedford, MA, USA). The UV spectra were pretreated prior to the multivariate data analysis in the following way: Linear baseline correction followed by a Savitzky–Golay smoothing (8 points, symmetric, 2nd polynomial order). The principal component analysis (PCA) models were calculated with mean centering, cross-validation, and the NIPALS algorithm. A partial least squares regression (PLS-R) model for the sugar concentrations was created with mean centering, segmented cross-validation according to the sample type, and the Kernel-algorithm. All cotton samples of each concentration were used to develop the PLS-R model. The PLS-R model was tested by predicting the honeydew content on the real cotton samples. Three different areas from these real cotton samples were also investigated by PCA with the aforementioned settings. To show the quality of the model, PCA was combined with quadratic discriminant analysis (QDA, 4 PCs). A fourth area was predicted by the PCA-QDA model.

MATLAB (MATLAB 9.2.0, Mathworks, MA, USA) and PLS_Toolbox (PLS Toolbox 8.5.1, Eigenvector Research, Inc., Wenatchee, WA, USA) were used for presenting the data.

## 3. Results and Discussion

### 3.1. Cotton Samples Impregnated with Sugar

Cotton samples were investigated using hyperspectral imaging in the UV region (225 nm–410 nm). In total, 21 samples were measured. The reference sample set was created to obtain a proper model to predict the honeydew content on real cotton samples. Figure 4a shows the averaged absorbance spectra in terms of reflectance. In general, the spectral shapes of all samples are quite similar. The most dominant band is pronounced approximately at 332 nm and a weak shoulder can be recognized at 346 nm (sh). An absorption band is observed at 261 nm corresponding to DNA absorbance [44]. Two weak shoulders are remarked at 291 nm, which can be assigned to the presence of protein and amino acids [44]. Despite the efficiency of the detector and the weak intensity light source in the spectral region between 250 and 270 nm [5], the signal is less intense, but nevertheless contains useful spectroscopic information for the actual application. A proof of the remaining performance of the setup in this range is given in Appendix A.

Figure 4b,c present the PCA model of the cotton samples at each sample pixel with different concentrations of sugar. Figure 4b shows the scores plot for the first (68.0%), second (22.0%), and fourth (1.0%) principal components (PCs). These PCs explain nearly 91.0% of the total variance. The variance on PC3 is not necessary to distinguish between different sugar concentrations. For completeness, PC3 is displayed in Appendix A. Different sugar concentrations on cotton can clearly be distinguished by the PCA scores. The mechanically cleaned sample (CLN) is separated on PC2. PC4 shows the separation between the highest sugar concentration and the lowest concentration. Slight overlapping is observed due to inhomogeneity in the impregnation procedure for the samples with sugar. The overlap tendency increases from higher to lower concentrations, and the variance within a sample increases with the concentration. Each cluster overlaps with the two closest sugar concentrations (higher and lower).

Figure 4c shows the loadings plot for PC1, PC2, and PC4. The strongest influence on PC1 is at 330 nm. Most of PC1 describes the morphology of the fiber itself. PC2 has a minimum at 280 nm and a maximum at 380 nm. These bands are responsible for the separation of the CLN sample from the others and distinguish between the different concentrations. For PC4, a maximum contribution is observed at 285 nm. These bands can be assigned to the presence of protein (Appendix A) [2]. The most significant differences between those loadings are found in the spectral region between 290 nm and 380 nm.

In order to validate the quality of the PCA model (Figure 4), the scores of the first four PCs were used to calculate a QDA. Table 4 presents the confusion matrix for the samples sprayed by sugar solutions. A confusion matrix yields a description of the capacity of the classification model [5]. An overall accuracy of 78.3% for the spectra is reached. Again, each type of sample is classified with the two closest sugar concentrations. The dark highlighted diagonal elements in Table 4 represent the accordance of the predicted and actual values. The light highlighted diagonal elements present the mispredicted spectra.

PLS-R was used for quantitative spectroscopic analysis. A PLS-R model was developed with a calibration sample set *n* = 21 to correlate the spectral information with the sugar concentration. Cotton samples (Table 2) were used for testing the performance of the model with a segmented cross-validation according to the sample type.

The variance explained by the model for the X- and Y-variables was 84% by using five factors. Accordingly, the five PLS factors were sufficient to describe the correlation between the spectra and the sugar content. The accuracy of the calibration and validation were evaluated using the coefficient of determination (*R*^2^) for the calibration (*R*^2^_c_ = 0.84) and segmented cross-validation (*R*^2^_cv_ = 0.80) model. The root-mean-square error of calibration (RMSEC = 0.009 g) and cross-validation (RMSECV = 0.01 g) indicate the model performance. A high *R*^2^_c_ and *R*^2^_cv_ are achieved with an extremely low RMSEC and RMSECV.

Figure 5 shows the PLS-R model for the cotton soaked in different concentrations of sugar in the UV region (200 nm–380 nm). Figure 5a presents the correlation between the reference vs. predicted, while the regression coefficients for the five-factor model are shown in Figure 5b. For model building and understanding the PLS-R factor loadings, loading weights for all five factors are displayed in the Appendix A. Sample E and sample F have similar ratios 0.0326 and 0.0322 (sugar/g per dried cotton/g), respectively, due to the preparation procedure’s limit. Therefore, they are overlapping in the reference vs. predicted plot. A negative band at 263 nm and a positive band at 284 nm can be assigned to protein absorbance. An average spectrum of pure dried protein is shown in the Appendix A. The protein information is pronounced in the spectra and mandatory for the model, even though the illumination and detector should be optimized [5]. From 300 nm to 400 nm, several features are observed that cannot be related to a common reason.

### 3.2. Predicting the Amount of Sugar and Honeydew Based on the Sugar PLS-R Model

The performance of the PLS-R model was tested by two methods. First, cleaned cotton samples were manually sprayed with aforementioned sugar concentrations to obtain a distribution of sugar droplets on the cotton surfaces. One benefit of hyperspectral imaging is to obtain the lateral information. Therefore, the PLS-R model was used to predict the sugar content on the different samples, and the result is shown in Figure 6. In the distribution map, a clear lateral classification of the different sugar concentrations resulting in different ratios of sugar/g per dried cotton/g was achieved. From sample A to sample F, the ratios decrease, and CLN samples are not soaked in sugar. Each sample was prepared three times, which are shown in the rows. Again, sample E and sample F are indistinguishable, due to the preparation procedure’s limit. Overall, the averaged predicted ratios decrease from samples A to CLN. The lateral inhomogeneities become visible through hyperspectral imaging.

Second, the PLS-R model was used to predict the honeydew content for each pixel of the real cotton types labeled light, strong, and very strong. In Figure 7, the resulting distribution maps are shown. As described in Figure 6, the distribution map shows a clear lateral classification of different ratios of sugar/g per dried cotton/g. The amount of sugar highly correlates with the amount of honeydew. Honeydew consists of different types of sugars and proteins [44]. From the very strong samples to the light samples, the ratios decrease. Three samples per type were collected and given in the rows.

Therefore, in Figure 7, red and blue pixels represent high and low concentrations of honeydew, respectively. As expected, the light sample displays a low honeydew concentration, while the other two samples show an increase in the laterally resolved honeydew concentration. Compared to the samples shown in Figure 6, the real samples show a more heterogeneous distribution of honeydew on the samples. Even in the strong and very strong samples, regions can be found where almost no honeydew is present. This can be seen in the presence of blue pixels on the distribution map of the very strong and strong samples.

## 4. Conclusions

In summary, this proof of principle study successfully demonstrated the identification and quantification of honeydew on real cotton samples by combining UV hyperspectral imaging (225 nm–410 nm) with multivariate data analysis. For this novel approach, a reference sample set was created based on mechanically cleaned cotton, which was impregnated with honeydew typical sugar and protein solutions for further UV hyperspectral imaging investigations. The PCA model enabled classification of the cotton samples according to their sugar concentration. A PLS-R model was created that was able to predict laterally resolved sugar/honeydew content pixel by pixel. This was shown for reference samples and for real cotton samples that were labeled as light, strong, and very strong contaminated by honeydew. The lateral distribution of the ratio of sugar/g per dried cotton/g per pixel gives a deeper insight into the distribution of honeydew on real cotton samples.

To the best of our knowledge, this is the first scientific work reporting the identification, quantification, and distribution of the amount of honeydew content by UV hyperspectral imaging. This approach may provide an advantage in the industrial environment in the practical process application and commercialization in the future. It enables control of the honeydew contamination in the industrial processing of cotton fibers in real time. Hence, each cotton batch, independent of the honeydew amount, can be manufactured to minimize waste and costs.

## Figures and Tables

**Figure 1 sensors-23-00319-f001:**
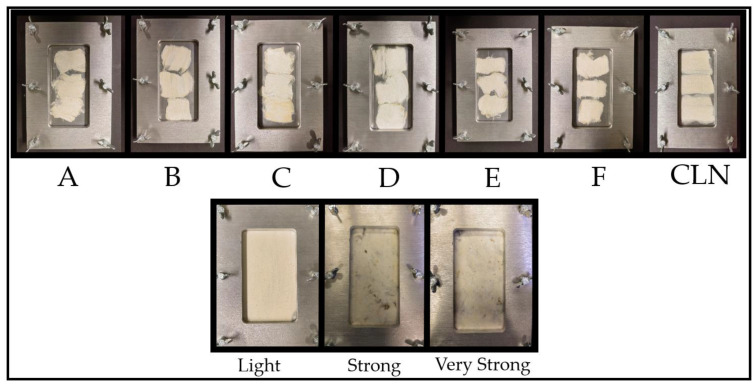
Overview of the samples pressed in the sample holder. For each concentration, three samples were prepared and measured at once ((**A**–**F**) and (**CLN**). Real cotton samples with different honeydew contents (light, strong, and very strong).

**Figure 2 sensors-23-00319-f002:**
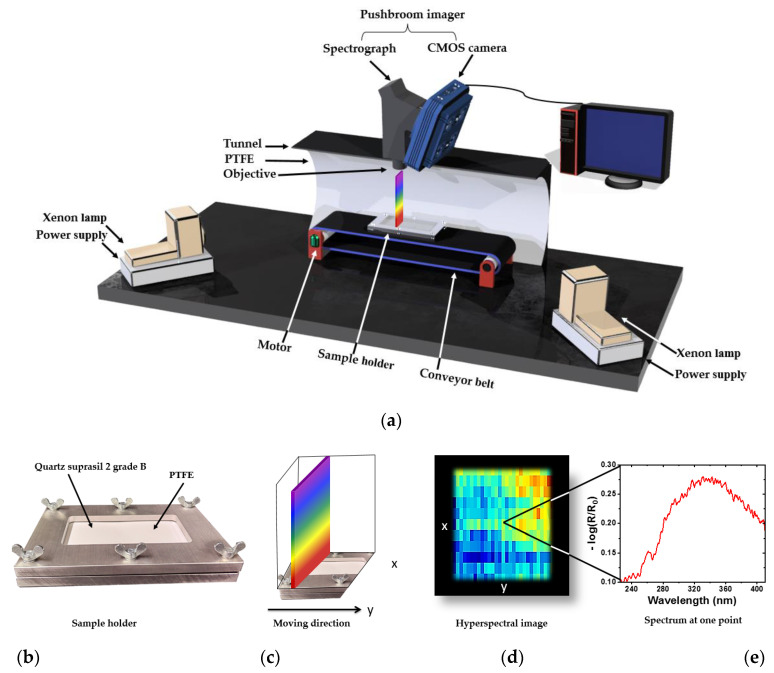
(**a**) Setup of a hyperspectral imaging system based on the pushbroom concept (the tunnel in the scheme was cut to show the inside). (**b**) Custom-made sample holder consisting of quartz glass as sample cover and PTFE as reference. (**c**) Pushbroom imager scanning principle. (**d**) Hyperspectral image generated immediately from the scanning of a sample. (**e**) UV spectrum after preprocessing for one point extracted from the image given in (**d**).

**Figure 3 sensors-23-00319-f003:**
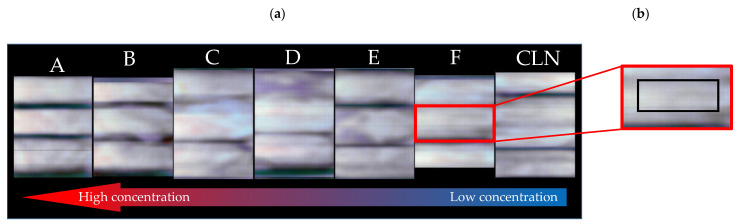
Example of data extraction. (**a**) Hyperspectral raw images at 290 nm of 18 cotton samples soaked with different concentrations of sugar (A highest to F lowest) and three cleaned cotton samples (CLN). For model building, all spectra were extracted manually. (**b**) Zoom-in-image of a cotton sample with the region of interest marked by a black rectangle.

**Figure 4 sensors-23-00319-f004:**
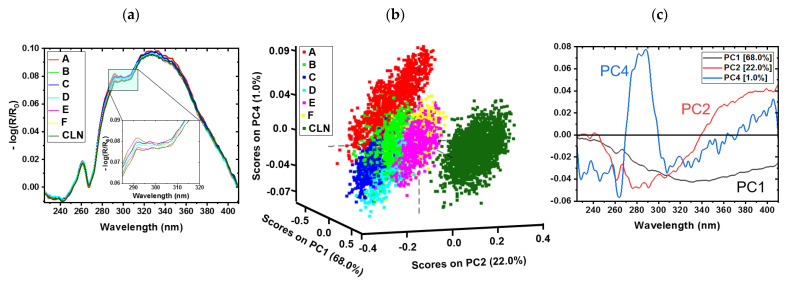
(**a**) Averaged UV spectra of cotton samples with sugar solutions in different concentrations: A (2 wt%, red), B (1 wt%, light green), C (0.5 wt%, blue), D (0.25 wt%, light blue), E (0.0125 wt%, pink), F (0.0625 wt%, yellow), and CLN (mechanically cleaned, dark green). PCA sugar model for the cotton samples with (**b**) scores and (**c**) corresponding loadings (PC1 black, PC2 red, and PC4 blue).

**Figure 5 sensors-23-00319-f005:**
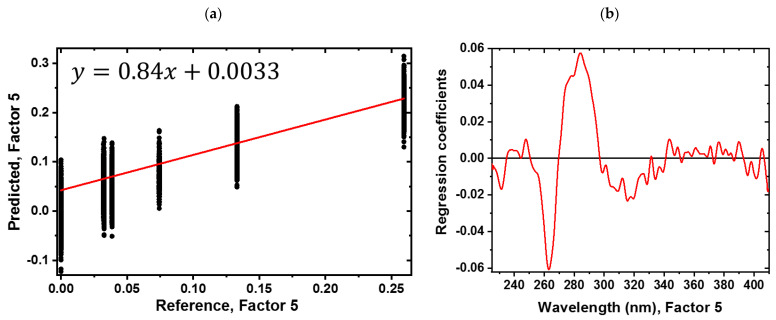
PLS-R model for different sugar concentrations in the UV region (225 nm–410 nm). (**a**) Predicted vs. reference plot with the equation of the regression function and (**b**) corresponding regression coefficients for the sugar content with a five factor PLS-R model.

**Figure 6 sensors-23-00319-f006:**
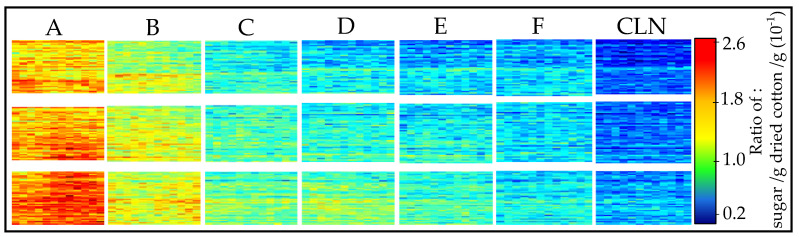
Distribution maps of the sugar content predicted on the mechanically cleaned cotton samples, which are manually sprayed by sugar solution. The prediction of each pixel is based on the PLS-R sugar model. Each rectangle represents a single cotton sample: (**A**) (2 wt%), (**B**) (1 wt%), (**C**) (0. 5 wt%), (**D**) (0. 25 wt%), (**E**) (0.125 wt%), and (**F**) (0.0625 wt%) and (**CLN**) (mechanically cleaned). The colored pixels (see the score value range) represent the sugar content, from low (blue) to high (red).

**Figure 7 sensors-23-00319-f007:**
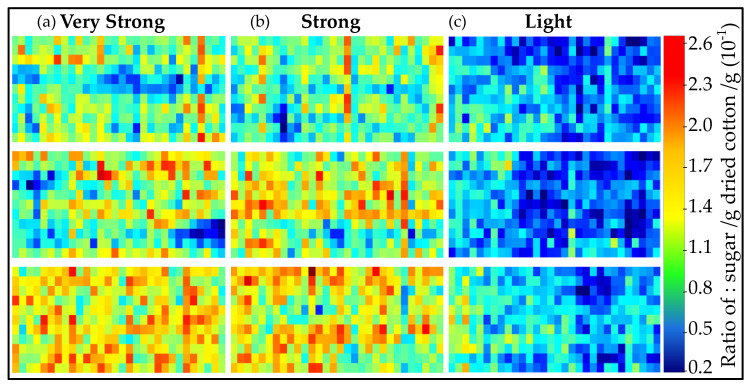
Distribution maps of the sugar content predicted on the real cotton samples, which are contaminated by honeydew. The prediction of each pixel is based on the PLS-R sugar model. Each rectangle represents a single cotton sample ((**a**) very strong, (**b**) strong, and (**c**) light). The colored pixels (see the score value range) represent the sugar content, from low (blue) to high (red).

**Table 1 sensors-23-00319-t001:** Description of the macronutrients and natural materials.

Macronutrients and Natural Materials	Samples	Description	Manufacture	CAS Number
1	Glucose	D-Glucose anhydrous Laboratory reagent grade	Fisher Scientific GmbH, Leics, UK	50-99-7
2	Fructose	D-Fructose, 99.0%	ThermoFisher GmbH, Kandel, Germany	57-48-7
3	Sucrose	D-Sucrose, ≥99.9% For Molecular Biology	Fisher Scientific GmbH, Fair Lawn, NJ, USA	57-50-1
4	Melezitose	D-(+)-Melezitose monohydrate, ≥99.0%	Sigma-Aldric Chemie GmbH, Steinheim, Germany	10030-67-8
5	Trehalose	D- Trehalose anhydrous, 99.0%	Acros Organics, Fair Lawn, NJ, USA	99-20-7
6	Protein	Bovine Serum Albumin (BSA) fraction V, lyophilized powder	PAN-Biotech GmbH, Aidenbach, Germany	9048-46-8

**Table 2 sensors-23-00319-t002:** The concentration of the sugar solutions and the weighted averaged sugar amount on cotton samples.

Sample Type	Sugar Concentration/wt%	Ratio of: Sugar/g Dried Cotton/g
A	2	0.2593
B	1	0.1331
C	0.5	0.0743
D	0.25	0.0386
E	0.125	0.0326
F	0.0625	0.0322
CLN	-	-

**Table 3 sensors-23-00319-t003:** The number of honeydew stickiness points on cotton samples.

Stickiness Type	Single Measurements	Average Number of Sticky Points	Sample
Light	2, 11, 5	6	4301
Strong	47, 45, 47	46	Sudan Girba Acala 3SG
Very strong	60, 69, 80	70	Sudan Gezira Acala type 3SG

**Table 4 sensors-23-00319-t004:** Confusion matrix of the PCA-QDA model with 4 PCs projected (overall accuracy 78.3%).

Actual								
**Predicted**	**Samples**	**A**	**B**	**C**	**D**	**E**	**F**	**CLN**
	A	818	33	1	0	1	1	0
	B	33	554	93	42	15	45	0
	C	10	148	405	72	0	33	0
	D	0	69	66	642	34	143	0
	E	0	6	0	36	664	290	0
	F	3	54	11	72	149	352	5
	CLN	0	0	0	0	1	0	1819

## Data Availability

The raw/processed data required to reproduce these findings cannot be shared at this time, as the data also form part of an ongoing Ph.D. thesis.

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
