# Peer review of "Prediction of Honeydew Contaminations on Cotton Samples by In-Line UV Hyperspectral Imaging"

_sensors, 2022, doi:10.3390/s23010319_

Round 1

Reviewer 1 Report

Dear Editor

The manuscript entitled “Predicting of Honeydew Contamination on Cotton Samples by In-Line UV Hyperspectral Imaging” involves valuable and novel approach for studying Honeydew contamination on cotton using the In-Line technology of UV imaging.  The study is almost comprehensive clear and well presented. Therefore, I recommend to be published in Remote Sensing after addressing few minor comments below.

-          Delete the world see when it comes in the form (see Table x).

-          I prefer to avoid using the subjective pronouns (I, we and our) in the scientific writing and replace it with passive voice.

-          It is not quite clear to me about how did the data matrix was constructed and what were the dimensions of the matrix when analyzed by PCA. (please include this in the manuscript).

-          According to the manuscript 21 samples were measures (line 220), but in the PCA scores plot (Figure 4 b) it is quite clear that number of samples is way more than 21 samples, please clarify this point in the discussion.

-          Why did the authors choose to illustrate the PCA results in a 3D scores model (using 3 PCs)?

In general, PCA is preferred to be presented in 2D models (using 2 PCs) to ease the prediction and avoid miss prediction especially that the 3D PCA is rotatable.

-          In lines 2445-2447, the authors said that each cluster overlaps with two closest sugar concentrations…..However, in Figure 4 b, I can see more than that, It would be better if correct clustering % were provided for each class of samples.

-           Line 269 sample E and sample F are not assigned in Figure 5.

-          In PLS-R, Figure 5a, please provide the equation of the straight line.

-          In the conclusion, line 327. The statement is not accurate concerning the PCA part (see above) and should be modified.

Author Response

Dear Reviewer,

Thank you very much, we discussed your points in the attached file.

Kind regards,

Mohammad Al Ktash

Reviewer 2 Report

The paper presents a method for detection and quantification of honeydew on cotton based on hyperspectral imaging (HSI) in the UV. The authors are right, HIS in the UV region is not often seen and is an interesting contribution. The proof-of-concept study seems to obtain reasonable and promising results, but there are parts of the manuscript that should be revised, and the procedure of modelling needs to be improved.

Line 37 HIS is not a combination of video and spectroscopy, it is a combination of imaging and spectroscopy. So remove “video”.

Sentence on line 41-42 is not clear – rewrite.

Line 83-84 PCA does not necessarily reveal the most “relevant” information. PCA decomposes the data into the principal components, whether they are relevant or not. For instance, in NIR spectroscopy, the fist and larges PC is often not relevant with respect to chemistry. PCA is purely mathematical and not automatically connected to relevance.

Lines 83-87 belong to the section Materials & methods.

Lines 89-95 belong to the section Materials & methods.

Lines 109-110 Sentence is unclear. Rewrite or remove.

Figure 3 a: Images at which wavelength are shown? What do the colors in the images represent (grey – reddish)?

Calibration and validation results are very similar. This is most likely due to overfitting – or insufficient validation. You use full cross-validation, but remember that your three samples within each group A, B, C etc can be regarded as similar and not as independent biological samples. For a bit more realistic validation, you should use segmented cross validation where all 3 samples within each group is taken out simultaneously. This is easy to do in Unscrambler.

It is not clear if you use a mean spectrum per sample or all pixel spectra in the regression model. Anyway: segmented cross validation, where A spectra, B spectra etc are left out at the same time is a more correct way to validate.

Lines 219-221 This is already explained in M&M and can be omitted from the Result part.

Line 222 What kind of baseline correction was used?

Lines 223-229 An interpretation of the absorption bands should be given. What molecules are absorbing?

Figure 6: This shows the cross validated predictions because this is the calibration set? Remember to update this figure with a more conservative modelling, where A samples are predicted based on B, C, D, E, F and CLN, B samples are predicted by A, C, D, E, F and CLN, etc

Line 306: “strong and light samples” Please be more precise in description.

Conclusion should not be an abstract or a repetition of the above. Suggest to remove first section (lines 324-333).

Language: Be consequent in the use of present and past tense.

Author Response

(The authors gave the same response as above.)

Reviewer 3 Report

The manuscript describes the use of hyperspectral imaging for the quantification of sugar and protein on cotton samples. The motivation for the research is the difficulty in processing cotton contaminated by honeydew, the natural excretion of insects. It seems that the quantification of contamination is the advance over previous work with hyperspectral images, which the authors state is able to discriminate between different sugars. However, it is not clear to me how well the mixture of sugars and BSA protein represent actual honeydew, or indeed the levels that are encountered in practice in the textile industry.

Principal components analysis shows that there are differences between the levels of sugar/protein mixtures and lines 211-214 mention PCA-QDA being used to predict sugar levels, but I can see no results from this analysis. PLS-R is also used to predict sugar/protein concentrations, but the predictions do not look very impressive. Although there is some correlation between the reference and predicted levels in Figure 5, there is a great deal of overlap with even a few of the highest reference levels being predicted as the lowest level. This figure suggests that only the fifth PLS-R component is used for prediction. Is this correct? Or is it, as would be more reasonable, that the first 5 components are used? In any case how useful are these results? There is very little explanation of the information in Table 3. In particular, what do the numbers for single measurements represent?

The PLS-R prediction of real honeydew levels shows little difference visually between “strong” and “very strong” but does show a difference between these and “light” contamination. Is this enough to be useful to the cotton industry? Some discussion of what would be acceptable and what would be rejected by quality control is necessary. Also, can these effectively visual results be converted by some method of combination to give an overall score for each sample that could actually be used for quality control?

As it is, this feels like a very preliminary study and I think it needs more work before it can be considered for publication.

Author Response

(The authors gave the same response as above.)

Round 2

Reviewer 3 Report

The revised version is improved and invludes information that was missing in the original version. The English language is also improved.